# Resolving ozone vertical gradients in air quality models

Katherine R. Travis<sup>1</sup>, Daniel J. Jacob<sup>1,2</sup>, Christoph A. Keller<sup>3,4</sup>, Shi Kuang<sup>5</sup>, Jintai Lin<sup>6</sup>, Michael J. Newchurch<sup>7</sup>, Anne M. Thompson<sup>8</sup>

<sup>1</sup>School of Engineering and Applied Sciences, Harvard University, Cambridge, MA, USA

5 <sup>2</sup>Department of Earth and Planetary Sciences, Harvard University, Cambridge, MA, USA

<sup>3</sup>Universities Space Research Association, Columbia, MD, USA
 <sup>4</sup>NASA Goddard Space Flight Center, Greenbelt, MD, USA
 <sup>5</sup>Earth System Science Center, University of Alabama in Huntsville, Huntsville, AL 35805, USA
 <sup>6</sup>Laboratory for Climate and Ocean-Atmosphere Studies, Department of Atmospheric and Oceanic Sciences, School of Physics,

Peking University, Beijing 100871, China
 <sup>7</sup>Atmospheric Science Department, University of Alabama in Huntsville, Huntsville, AL 35805, USA
 <sup>8</sup>Earth Science Division, NASA Goddard Space Flight Center, Greenbelt, MD 20771, USA

Correspondence to: Katherine R. Travis (ktravis@fas.harvard.edu)

Abstract. Models severely overestimate surface ozone in the Southeast US during summertime and this overestimation has
implications for the design of air quality regulations. We use the GEOS-Chem model to interpret ozone observations from aircraft (SEAC<sup>4</sup>RS), ozonesondes (SEACIONS), and surface sites (CASTNET) in August-September 2013. After correcting for a 30-50 % NO<sub>x</sub> emission overestimate in the US EPA National Emission Inventory, we find that the model is unbiased relative to aircraft observations below 1 km. However, surface observations of maximum daily 8-h average (MDA8) ozone are still biased high in the model (averaging 48 ± 9 ppb) compared to observations (40 ± 9 ppb). The low tail in the observations (MDA8 ozone < 25 ppb)</li>

- is associated with rain and is not captured by the model. The model bias decreases by 3 ppb when accounting for the subgrid vertical gradient between the lowest model level (centered 60 m above ground) and the measurement altitude (10 m). The model underestimates low cloud cover, but this underestimate is insufficient to explain the remaining surface ozone bias because the response of model ozone to cloud cover is weaker than observed. Midday ozonesondes at Huntsville, Alabama show mean decreases in ozone from 1 km to the surface of 4 ppb under clear-sky and 7 ppb under low cloud, whereas the model decreases by
- only 1 ppb under both conditions. By contrast, potential temperature below 1 km is well-mixed in both the observations and the model. The observations thus imply a strong asymmetry between top-down and bottom-up mixing that is missing from GEOS-Chem and appears to be insufficiently represented in current air quality models. A sensitivity simulation reducing top-down eddy diffusion and removing top-down non-local vertical transport of ozone can reproduce the observed ozone gradients in the mixed layer.

## 30 1 Introduction

Ground-level ozone is harmful to human health and vegetation. Ozone is produced in the troposphere when volatile organic compounds (VOCs) and carbon monoxide (CO) are photochemically oxidized in the presence of nitrogen oxide radicals (NO<sub>x</sub>  $\equiv$  NO+NO<sub>2</sub>). Natural sources of VOCs, CO, and NO<sub>x</sub> from the biosphere, wildfires, and lightning contribute an ozone background. Anthropogenic sources, mainly from fuel combustion, increase ozone levels. The chemistry involved is complex and non-linear.

Air pollution control strategies rely on chemical transport models (CTMs) to identify the most effective emission reductions, but confidence in these models can be limited by their inability to reproduce ozone observations. The Southeast US in summer is a particularly problematic region, as models tend to greatly overestimate surface ozone levels (Lin et al., 2008; Fiore et al., 2009; Reidmiller et al., 2009; Chai et al., 2013; Brown-Steiner et al., 2015; Canty et al., 2015; Travis et al., 2016; Lin et al., 2017). An

intercomparison of 21 models by Fiore et al. (2009) showed an average overestimate of 25 ppb in the Southeast in August. Here we use a combination of aircraft, ozonesonde, and surface observations in summer 2013 to better understand this overestimate and draw general insights for ozone air quality modeling.

- 5 The Southeast US in summer is characterized by relatively high NO<sub>x</sub> emissions, very high emissions of biogenic isoprene, strong insolation, and frequent regional stagnation, all conditions favorable for producing elevated ozone. A range of explanations have been proposed for the model overestimates of ozone in that region including excessive ozone background over the Gulf of Mexico (Fiore et al., 2003), uncertainty in isoprene emissions and chemistry (Fiore et al., 2005; Horowitz et al., 2007; Squire et al., 2015), insufficient ozone dry deposition (Lin et al., 2008), missing halogen chemistry (McDonald-Buller et al., 2011), and excessive NO<sub>x</sub>
- 10 emissions in current inventories (Travis et al., 2016).

Detailed probing of the chemical environment of the Southeast US took place in summer 2013 with surface and aircraft observations from the Southeast Atmosphere Studies (SAS) in June-July (Carlton et al., 2016), the NASA SEAC<sup>4</sup>RS aircraft campaign in August-September (Toon et al., 2016), and the SEACIONS ozonesonde network
(https://tropo.gsfc.nasa.gov/seacions/), adding to the long-term ozone air quality monitoring network. In previous work by Travis et al. (2016), we applied the GEOS-Chem CTM at 0.25° × 0.3125° spatial resolution to the simulation of SEAC<sup>4</sup>RS observations. The standard model overestimated ozone by 12 ppb below 1.5 km altitude. On the basis of observations of NO<sub>x</sub> and its oxidation

- products, together with national nitrate wet deposition data, we showed that the NO<sub>x</sub> National Emission Inventory (NEI) from the US Environmental Agency (EPA, 2015) was too high by 30-50 %. This finding was subsequently supported by SAS observations
  (Miller et al., 2017). Previous studies had documented such a NO<sub>x</sub> NEI bias in urban areas (Fujita et al., 2012; Yu et al., 2012; Brioude et al., 2013; Anderson et al., 2014), but our results suggest that the bias is national in extent. After correcting this NO<sub>x</sub>
- emission overestimate in GEOS-Chem, we found that we could match the SEAC<sup>4</sup>RS aircraft observations below 1.5 km altitude, but the model mean bias against surface network observations was still  $6 \pm 14$  ppb. Midday ozonesonde observations showed an increase of ozone with altitude in the lowest 1 km of the atmosphere that the model failed to capture. Here we examine the origin
- of this ozone vertical gradient and the implications for modeling surface ozone.

### 2 GEOS-Chem simulation

The GEOS-Chem simulation used here is as described by Travis et al. (2016). It is based on GEOS-Chem version 9.02 with detailed oxidant-aerosol chemistry (<u>www.geos-chem.org</u>) and is driven by assimilated meteorological data from the Goddard Earth Observing System – Forward Processing (GEOS-FP) product of the NASA Global Modeling and Assimilation Office (GMAO)

- using the GEOS-5.11.0 general circulation model (GCM). The GEOS-FP data have a native horizontal resolution of 0.25° latitude by 0.3125° longitude, with 72 levels in the vertical and a temporal resolution of 3 h (1 h for surface variables and mixing depths). This native 0.25° × 0.3125° horizontal resolution is used in GEOS-Chem over North America and adjacent oceans (130° 60° W, 9.75° 60° N), with boundary conditions from a global simulation with 4° × 5° horizontal resolution.
- The model representations of planetary boundary layer (PBL) mixing and ozone deposition are particularly relevant for this work. The PBL is defined as the column of air in contact with the surface on a daily basis. Observations show that the PBL over the Southeast US in summer extends to 1-3 km altitude and is capped by a semi-permanent subsidence inversion (Toon et al., 2016). Within the PBL, the unstable mixed layer driven by surface heating rises rapidly in the morning to reach a maximum altitude

(mixing depth) of  $1.7 \pm 0.4$  km by afternoon, as observed in SEAC<sup>4</sup>RS by aerosol lidar (Zhu et al., 2016), before collapsing in the evening. The afternoon mixed layer is often capped by shallow fair-weather cumuli (cloud convective layer) constituting the upper part of the PBL. The PBL is resolved in the GEOS-FP data (and hence in GEOS-Chem) by 18 vertical levels below 3 km, and 8 below 1 km of approximately equal thickness. The lowest level is centered at 60 m above ground. Turbulence in the mixed layer

follows a clear-sky non-local parameterization from Holtslag and Boville (1993), as implemented in GEOS-Chem by Lin and McElroy (2010). The parameterization uses mixing depths from the GEOS-FP data, which are diagnosed as the GCM model level above which the eddy diffusivity for heat ( $K_h$ ) falls below a threshold value of 2 m<sup>2</sup> s<sup>-1</sup> (McGrath-Spangler and Molod, 2014). GEOS-FP mixing depths were found to be 40 % too high compared to the SEAC<sup>4</sup>RS aerosol lidar data and this was corrected in the GEOS-Chem simulations (Zhu et al., 2016). Additional turbulence due to cloud cooling at the PBL top is included in the GEOS-

5.11.0 GCM following Lock et al. (2000) but not in the Holtslag and Boville (1993) scheme.

Ozone deposition in GEOS-Chem follows the resistance-in-series scheme of Wesely (1989) as implemented by Wang et al. (1998) and further modified for SEAC<sup>4</sup>RS conditions by Travis et al. (2016). The mean daytime (09:00-16:00 local) ozone deposition velocity over the Southeast US in the model is  $0.7 \pm 0.3$  cm s<sup>-1</sup> during August-September 2013. Comparison with ozone deposition

- measurements by Finkelstein et al. (2000) at Duke Forest, North Carolina shows good agreement with a mean ozone deposition velocity of 0.8 cm s<sup>-1</sup> during daytime. Aircraft eddy covariance flux measurements over the Ozarks forest during SEAC<sup>4</sup>RS indicate a daytime ozone deposition velocity of  $0.8 \pm 0.1$  cm s<sup>-1</sup>, in agreement with the local GEOS-Chem value of 0.9 cm s<sup>-1</sup> (Wolfe et al., 2015).
- Detailed evaluations of GEOS-Chem with SAS and SEAC<sup>4</sup>RS observations have been reported in previous studies. Initial evaluations led to corrections of daytime mixing depths (Zhu et al., 2016), NEI NO<sub>x</sub> emissions (Travis et al., 2016), and isoprene chemistry (Fisher et al., 2016; Travis et al., 2016). After these corrections, the model was found to be successful in reproducing surface and aircraft observations of aerosol composition (Kim et al., 2015b; Marais et al., 2016) and organic nitrates (Fisher et al., 2016), and aircraft observations of formaldehyde (Zhu et al., 2016), glyoxal (Miller et al., 2017), and ozone and its precursors
- (Travis et al., 2016; Yu et al., 2016). Travis et al. (2016) presented model comparisons to observations of (1)  $NO_x$ , (2) the relationship of ozone to  $NO_x$  oxidation products (a measure of the ozone production efficiency), and (3) isoprene nitrates and peroxides tracking the high-NO (ozone-producing) and low-NO pathways for isoprene oxidation. This evaluation lends some confidence in the model simulation of ozone chemistry.

## **3** Ozone frequency distributions in the mixed layer and surface air

Figure 1 (left panel) shows the frequency distribution of afternoon (12-18 local time) ozone concentrations in August-September 2013 measured by the SEAC<sup>4</sup>RS DC-8 aircraft in the mixed layer at 0.4-1.0 km altitude. The mean ozone in the mixed layer as measured by the aircraft is 50 ± 10 ppb. The model sampled along the aircraft tracks is in good agreement (52 ± 10 ppb, *r*=0.54). The model does not capture the observed extremes and this can be simply explained by numerical diffusion (Yu et al., 2016). In particular, observations above 75 ppb are associated with urban (Houston) and agricultural fire plumes (Travis et al., 2016).

Also shown in Figure 1 (right panel) is the frequency distribution of maximum daily 8-hour average (MDA8) ozone at the CASTNET surface network for the same period (https://www.epa.gov/castnet). CASTNET monitors air quality in rural areas and is therefore representative of regional air quality. The mean MDA8 ozone measured at CASTNET sites is  $40 \pm 9$  ppb, while the

<sup>35</sup> 

corresponding model mean is  $48 \pm 9$  ppb, for a high mean bias of  $8 \pm 9$  ppb. The model shows only a 4 ppb difference between the mixed layer sampled by the aircraft and the surface, but the observations imply a 10 ppb difference.

- Part of the surface bias in the model can be simply attributed to representation error. The lowest model grid-point in GEOS-Chem 5 is centered at 60 m above the local surface. The CASTNET measurements are typically at 10 m altitude. Implicit model ozone concentrations at 10 m can be inferred from the values at 60 m and the local ozone deposition velocity by applying the model aerodynamic resistance ( $R_a$ ) between 60 and 10 m. The formula for this correction is presented in Zhang et al. (2012). We combine a typical friction velocity  $u^* = 0.4$  cm s<sup>-1</sup>, daytime Monin-Obhukov length |L| = 40 m, and  $R_a = 0.07$  s cm<sup>-1</sup> with an ozone deposition velocity of 0.8 cm s<sup>-1</sup> and find an average ozone decrease of 3 ppb between 60 m and 10 m. The right panel of Figure 1 includes
- the implied model pdf at 10 m altitude, as inferred from the local model values of  $R_a$ ; the model mean is 45 ± 8 ppb. The mean bias relative to observations decreases to 5 ± 9 ppb. We apply this correction in all following model comparisons.

The relatively low surface ozone measured at CASTNET sites in August-September 2013 reflects lower-than-average but not anomalous conditions. Figure 2 (top panel) shows the long-term trend of August-September MDA8 ozone in the Southeast US

from 1987 to 2015. There is a 0.4 ppb a<sup>-1</sup> decrease due to emission controls (Cooper et al., 2012). The 2013 data are 2 ppb below the linear fit to that long-term trend, and this may be due to cooler and wetter conditions than average (bottom panel).

The frequency distribution of MDA8 ozone at the CASTNET sites in Figure 1 shows a population of very low ozone concentrations below 25 ppb that the model does not capture at all. Previous work has suggested that this population could be due to tropical air

transported from the Gulf of Mexico (Fiore et al., 2002; McDonald-Buller et al., 2011). However, we find that the observed occurrence of low values is distributed across the Southeast and is not related to distance from the Gulf. Four SEAC<sup>4</sup>RS flights sampled air over the Gulf of Mexico and showed a median ozone concentration of 26 ppb below 1.5 km with the model in close agreement (Travis et al., 2016). Rain may be an additional factor driving low ozone, as discussed below.

## 4 Relationship to cloud cover and precipitation

- We examined whether the 5 ± 9 ppb mean model bias in simulating MDA8 ozone at surface sites could be attributed to cloudy and rainy conditions. Such a bias would not affect the comparison to aircraft observations, which generally targeted clear-sky conditions. For this purpose we segregated the frequency distributions of ozone at CASTNET sites between clear-sky, low-cloud with no rain, and rainy days. Low cloud in the observations was diagnosed by 20-minute averaged data at nearby airports from the automated surface observing system network (ASOS) sensors collected by the Iowa Environmental Mesonet (IEM) with 371
- locations in the Southeast US (http://mesonet.agron.iastate.edu/request/download.phtml). Cloud data below 680 hPa are reported in oktas. Low-cloud conditions are defined here as average daytime cloud fraction greater than 3 oktas (3/8 cloud fraction), excluding rainy conditions, and clear-sky conditions are defined as less than 0.5 oktas (0.5/8 cloud fraction). Rainy conditions are defined by daily average rainfall exceeding 6 mm in the PRISM data regridded to 0.25° × 0.3125°. Rainy conditions in the model are diagnosed in the same way as in the observations, while cloudy conditions are diagnosed from cloud fractions at different

vertical levels below 680 hPa using the maximum random overlap scheme (MRAN) of Liu et al. (2006). In the remainder of this paper, "cloudy" conditions refer to low-cloud conditions.

- Figure 3 shows the segregated pdfs of surface ozone in the observations and the model. The days for a given sky condition are not 5 necessarily the same in the observations and the model. We see that ozone decreases from clear to low-cloud to rainy conditions in both the observations and the model. The model is heavily biased toward clear-sky. The average daytime low-cloud cover across the entire Southeast is  $32 \pm 9$  % from the ASOS sensors but only  $7 \pm 3$  % in the GEOS-FP data. The GEOS-5 GCM underlying the GEOS-FP data uses a critical RH to trigger cloud formation (Molod et al., 2012; Molod et al., 2015) and the cloud bias could result from the setting of this trigger (Naud et al., 2010). The low-cloud bias in GEOS-FP is also apparent in comparison to satellite
- observations from the Clouds and the Earth's Radiant Energy System (CERES) instruments (Minnis et al., 1995; Minnis et al., 2011). Figure 4 compares CERES low-cloud fractions in August-September 2013 in the Southeast with GEOS-FP values. The mean observed low-cloud fraction is 21 ± 4 % as compared to 9 ± 2 % in GEOS-FP. The mean in-cloud optical depth is 45 ± 3 in both CERES and GEOS-FP. Thus the optical depth of low clouds in GEOS-FP is consistent with observations but the cloud frequency is too small. Table 1 shows that the underestimate in GEOS-FP cloud fraction is mainly due to a lack of fair-weather
- cumulus. Climate models generally tend to underestimate low cloud cover (Zhang, 2005; Mueller et al., 2006; Chepfer et al., 2008; Naud et al., 2010; Kay et al., 2012; Nam et al., 2012). The GEOS-Chem underestimate of sulfate aerosol production in SEAC<sup>4</sup>RS, previously attributed by Kim et al. (2015) to a missing SO<sub>2</sub> oxidation pathway involving Criegee biradicals, could instead be due to insufficient cloud processing.
- We see from Figure 3 that the mean bias between model and observed surface ozone vanishes when only clear-sky conditions are considered, but persists under low-cloud and rainy conditions. Thus the bias cannot be simply attributed to insufficient cloud in the model. If we apply the observed frequencies of clear-sky, cloudy, and rainy days from Figure 3 to the model mean ozone concentrations for each category, we decrease the mean model MDA8 ozone bias at CASTNET sites by only 1 ppb. This is because of the weaker response in the model to cloud cover and rain (4 ppb relative to clear-sky) than observed (7 ppb and 11 ppb
- respectively). Kim et al. (2015a) previously observed a 1 ppb decrease in ozone per 10 % increase in cloud cover over the contiguous United States, and found that their model response to cloud (from the NOAA National Air Quality Forecast) was approximately half that, a similar bias to our model. We conducted a model sensitivity study with the low cloud fraction adjusted to the mean observed value of 32 % from the ASOS observations. This simulation perturbs model photolysis but does not modify other meteorological variables. We find an ozone decrease of only 1 ppb and thus photolysis appears to be only a minor effect.
- Previous urban-scale model studies have found larger cloud effects on surface ozone from changes in photolysis (Pour-Biazar et al., 2007;Tang et al., 2015), but larger-scale studies find a weaker effect consistent with our findings (Voulgarakis et al., 2009). SEAC<sup>4</sup>RS observations of actinic fluxes in SEAC<sup>4</sup>RS show cloud effects consistent with radiative transfer models (Ryu et al., 2017).
- The largest difference between model and observations occurs on rainy days. Rainy days account for over half of all days with observed MDA8 ozone below 25 ppb. Thus, the inability of the model to reproduce the low tail in the observed ozone distribution appears to be due in large part to positive bias on rainy days. This could reflect vertical stratification from surface evaporative cooling that is not properly captured in the model. The effect of precipitation on ozone through wet scavenging is negligible.

Rainfall or dew may also enhance the non-stomatal component of ozone dry deposition (Finkelstein et al., 2000; Altimir et al., 2006; Potier et al., 2017) but the mechanism for this enhancement is uncertain and is not included in the model.

#### 5 Ozone vertical profiles at Huntsville

The analysis above suggests that insufficient model response to cloud conditions and rain could be the cause of the remaining surface ozone bias. We examined whether this could be related to excessive vertical mixing in the model by using the SEACIONS ozonesonde data from Huntsville, Alabama (31 launches at 10-13 local time during August-September 2013; https://tropo.gsfc.nasa.gov/seacions/). The ozonesondes measure ozone at approximately 5-m resolution from the surface through the stratosphere but the 5-m resolution data are averaged and reported at coarser resolution to achieve reasonable noise statistics. We interpolate the data to the model vertical resolution (approximately 130 m) and down to 10 m above ground. Huntsville is a

10 small-sized city at 200-m ASL with forested land cover and little topography, and the ozonesonde data can be viewed as regionally representative (Newchurch et al., 2003).

The top panel of Figure 5 compares the time series of ozonesonde observations at Huntsville up to 12 km altitude to the corresponding GEOS-Chem values. The model successfully captures the large-scale features in the free troposphere above 3 km with no significant bias ( $1 \pm 14$  ppb). A comparison of the modeled and observed mean profile at Huntsville is shown in Travis et

15 with no significant bias (1  $\pm$  14 ppb). A comparison of the modeled and observed mean profile at Huntsville is shown in Travis et al. (2016).

The bottom panel of Figure 5 shows the ozonesonde vertical profiles with more resolution below 3 km. As for the CASTNET data, we infer model ozone at 10 m for each ozonesonde launch from the simulated concentration at the lowest model level (60 m) and local values of the aerodynamic resistance and ozone deposition flux. For the ensemble of ozonesonde launches, we find a mean 10-60 m aerodynamic resistance of 0.04 s cm<sup>-1</sup> and an ozone deposition velocity of 0.8 cm s<sup>-1</sup>, resulting in a mean model difference of  $1.6 \pm 0.5$  ppb ozone between 60 and 10 m. This is less than the mean 3 ppb effect found for MDA8 ozone at CASTNET sites (Section 3), because the MDA8 8-h averaging window includes periods with greater stability than midday. The implied model gradient at Huntsville is consistent with the mean observed difference of  $0.7 \pm 0.9$  ppb in the ozonesonde data between 60 and 10 m.

We find that surface (10 m) ozone at Huntsville shows similar behavior to the CASTNET network. Mean observed surface ozone from the ozonesondes ( $43 \pm 12$  ppb) compares well with the observed CASTNET MDA8 ozone shown in Figure 1. Ozone is lowest on rainy days (n=6,  $36 \pm 12$  ppb), diagnosed from the PRISM data, similar to our finding at CASTNET sites in Figure 3. The

- 30 lowest ozone (18 ppb) on September 21 occurred on the day with the most rainfall in the time series (50 mm), in air originating from the Gulf of Mexico. We do not find a significant difference in surface ozone at Huntsville between cloudy conditions (n=14, 43 ± 13 ppb) and clear conditions (n = 5, 44 ± 13 ppb), but this may be due to the small sample size. The modeled surface ozone for the ozonesonde launches is 48 ± 9 ppb and the mean model bias is 5 ± 9 ppb (r=0.67), same as at the CASTNET sites.
- 35 The mean ozone decrease from 1 km down to the surface is steeper in the observations (6 ± 5 ppb) than in GEOS-Chem (1 ± 3 ppb) and agrees well with the implied gradient shown in Figure 1 between the SEAC<sup>4</sup>RS aircraft and CASTNET surface observations. The mean observed decrease is 4 ± 5 ppb on clear days (*n*=5) and 7 ± 6 ppb on cloudy days (*n*=14) but this difference is not statistically significant (*p* = 0.2). The model decrease is less than 1 ppb on either clear (*n*=15) or cloudy (*n*=3) days. This

confirms that the model overestimate of surface ozone is due to underestimate of the gradient in the lowest km, particularly under cloudy conditions but also under clear-sky conditions.

## 6 Top-down PBL mixing of ozone

Figure 6 shows ozone and potential temperature profiles on two typical days where model and observations agree on the clear and low-cloud classification. These specific days have free tropospheric biases but our interest here is in the simulation of the PBL vertical gradient. On the clear-sky day (Sep 4), the model is well-mixed throughout the lowest km but the observations show a vertical gradient, particularly in the lowest 300 m. The potential temperature profile is well-mixed in both the observations and model. On the cloudy day (Aug 16) there is a steady gradient below 1 km in the observations that the model does not reproduce. The grey shading on Figure 6 shows the convective cloud layer in the upper part of the PBL and again the model does not capture

- the gradient in that layer. We conducted a sensitivity on-line simulation in the GEOS-5 GCM using the GEOS-Chem chemical module (Long et al., 2015) and including the GEOS-5 PBL mixing scheme of Lock et al. (2000), but found the same excessive downward mixing of ozone as in the off-line GEOS-Chem. The inconsistency between potential temperature, which is well-mixed in both the observations and the model, and ozone, for which the observations show a vertical gradient absent from the model, suggests a bottom-up vs. top-down asymmetry in vertical mixing that is missing from both the Holtslag and Boville (1993) and
- Lock et al. (2000) PBL schemes.

Wyngaard and Brost (1984) used large-eddy simulations to investigate top-down vs. bottom-up differences in eddy diffusion parameterizations of PBL mixing. They show that eddy diffusion coefficients ( $K_z$ ) for top-down transport should be about 60 % lower than for bottom-up transport, due to the role of surface-driven buoyant plumes in contributing to bottom-up transport.

Additional non-local vertical transport in PBL schemes, developed originally for heat flux, is mostly intended to resolve buoyant plumes (Deardorff, 1966; Holtslag and Moeng, 1991) and should be formulated differently for top-down transport (Xie and Fung, 2014). We conducted a sensitivity simulation for the two sample days of Figure 6 where the Holtslag and Boville (1993) mixing scheme was modified for ozone to decrease  $K_z$  by 60 % and remove the non-local term. As shown in Figure 6, this fully corrects the ozone gradient.

The need for asymmetric top-down vs. bottom-up PBL mixing for air quality applications has long been recognized (Pleim and Chang, 1992), and is presently implemented in the EPA Community Multiscale Air Quality (CMAQ) and in the Comprehensive Air Quality Model with Extensions (CAMx) using the Asymmetrical Convection Model version 2 (ACM2) (Pleim, 2007a, b). The ACM2 has the same eddy diffusion component as Holtslag and Boville (1993) but a different form of non-local parameterization.

It treats upward convective transport with a non-local buoyant component, but downward transport as a slower, layer-by-layer process. However, comparisons to ozonesonde and aircraft observations suggest that ACM2 still has excessive mixing for ozone down to the surface (Tang et al., 2011, Goldberg, 2015).

#### 7 Conclusions

Models overestimate summertime surface ozone in the Southeast US. We showed previously using the GEOS-Chem model that 35 this overestimate is due in part to an overestimate of NO<sub>x</sub> emissions in the US EPA National Emission Inventory (Travis et al., 2016). However, midday ozonesondes also show a large vertical gradient of decreasing ozone below 1 km altitude that is at odds

with the strong mixing expected from models. Here we investigated the cause of this discrepancy through the combined analysis of August-September 2013 ozone observations from aircraft (SEAC<sup>4</sup>RS), surface (CASTNET), and ozonesondes (SEACIONS).

Statistical comparison of the GEOS-Chem model to aircraft observations of ozone below 1 km shows no significant bias  $(50 \pm 10 \text{ ppb} \text{ observed}, 52 \pm 10 \text{ ppb} \text{ model})$ , but the maximum daily 8-h average (MDA8) surface ozone at CASTNET sites is overestimated by  $8 \pm 9 \text{ ppb} (40 \pm 9 \text{ ppb} \text{ observed}, 48 \pm 9 \text{ ppb} \text{ model})$ . The lowest model level is centered at 60 m above ground while the observations are at 10 m; thus a subgrid correction must be applied using the model aerodynamic resistance to dry deposition. This correction, which is generally ignored in models, averages 3 ppb in our case; it is relatively large because the MDA8 8-hour window can include convectively stable conditions. The resulting model ozone at 10 m is  $45 \pm 8$  ppb, still significantly higher than

10 observed. August-September 2013 was cooler and wetter than average but the effect on ozone was small, averaging 2 ppb at CASTNET sites. The low tail of observed MDA8 ozone (<25 ppb) was largely associated with rainy conditions.

The GEOS-FP meteorological data driving GEOS-Chem are biased toward clear-sky, and this bias would be expected to contribute to the overestimate of ozone. However, we find that the model MDA8 ozone is only 4 ppb lower under low-cloud and rainy

- conditions than in clear sky, whereas in the observations that difference is 7 ppb under low-cloud conditions and 11 ppb under rainy conditions. Midday ozonesonde data from Huntsville, Alabama show a 6 ppb decrease from 1 km to the surface (4 ppb under clear-sky, 7 ppb under low cloud), whereas the model shows only a 1 ppb decrease. Thus the model has excessive top-down mixing of ozone; this is seen using both the Holtslag and Boville (1993) PBL scheme in the off-line GEOS-Chem and the Lock et al. (2000) scheme in the GEOS-5 GCM. By contrast, potential temperature shows similar strong vertical mixing in the observations
- and the model. Bottom-up mixing (as for heat) is known to be faster than top-down mixing (as for ozone) because of buoyant plumes but the two above schemes do not include this asymmetry. The ACM2 scheme (Pleim, 2007a, b) includes this asymmetry, but previous evaluations suggest that this scheme still has excessive downward mixing of ozone. We find in a sensitivity simulation that decreasing top-down eddy diffusion following Wyngaard and Brost (1984) and suppressing top-down non-local vertical transport allows GEOS-Chem to successfully simulate the observed ozone gradient in the mixed layer. More work is needed to
- describe the top-down mixing of ozone for air quality applications.

### 8 Data availability

Cloud (ASOS) data from the Automated Surface Observing System can be downloaded here: http://mesonet.agron.iastate.edu/request/download.phtml. PRISM temperature and precipitation data can be downloaded here: http://www.prism.oregonstate.edu/historical/. The SEACIONS ozonesonde data be accessed here: can 30 https://tropo.gsfc.nasa.gov/seacions. The CERES cloud fraction and cloud optical depth observations are available at

http://doi.org/10.5067/Aqua/CERES/ISCCP-D2LIKE-MERG00\_L3.003. The SEAC<sup>4</sup>RS aircraft data can be found here: https://www-air.larc.nasa.gov/missions/seac4rs/DC8-Extract.html. CASTNET data are available at: https://www.epa.gov/castnet.

#### 9 Competing Interests

The authors declare that they have no conflict of interest.

Acknowledgements

We thank Randal Koster (NASA), Dan Goldberg (ANL), and Taylor Jones, Eloise Marais, Rachel Silvern, and Lu Shen (Harvard) for helpful discussions. This work was supported by the NASA Earth Science Division. AMT acknowledges SEACIONS support from the Tropospheric Chemistry Program to NASA/Goddard, NOAA/ESRL/GMD and originally to Penn State University (Grant NNX12AF05G).

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
