# Peer review of "Resolving ozone vertical gradients in air quality models"

_Atmospheric Chemistry and Physics, 2017_

## Referee Comment (RC1) · Anonymous Referee #2 · 30 Aug 2017

General comments

I think that the issue that this paper raises, the modeling of vertical gradients of ozone, is interesting and potentially important. However, there are too many shortcomings to the modeling to give a definitive analysis of the issue. An obvious shortcoming is that the lowest model level is 60 m while the measurements are at 10 m. Most other air quality models use much finer grid resolution near the surface. Furthermore, I strongly object to using different PBL schemes for meteorology and chemistry. Not only has the PBL height been reduced for the chemistry but the $K_z$ is also reduced and non-local term eliminated for the proposed correction. This is unjustified. If the same scheme does not give realistic results for both meteorology and chemistry, it should not be used for combined meteorology and air quality modeling. The other corrections and

adjustments, such as reducing NOx emissions, that are made in order to get better results also call into question the validity of the study and its findings.

Specific comments

P2 ln17-19: The statement: "we showed that the NOx National Emission Inventory (NEI) from the US Environmental Agency (EPA, 2015) was too high by 30-50 %." is much too strong. All they should say is that this model with its many demonstrated errors better predicts ozone and NOx with a 30-40% emission cut. I do not believe that it's been proven that the NEI is over predicting NOx emissions by this amount.

P3 ln6-7: I don't understand how the mixing depth (h) can be defined by Kh values when according to Holtslag and Boville (1993) the Kh is a function of h which is defined by bulk Richardson number. Also, if the mixing depth is 40% too high, then this problem should be diagnosed and fixed. It is not reasonable to "correct" this in Geos-Chem. The result would be inconstancies between chemical concentration profiles and meteorological profiles which would lead to many errors in the chemical simulation including incorrect advection, errors in temperature and humidity especially above the GEOS-Chem h but below the GEOS h. I think that such "corrections" in AQ modeling should not be acceptable practice.

P3 ln20-22: More explanation and justification should be provided for the corrections and adjustments to PBL, emissions, and chemistry. The reader should not have to look up these other papers to know what was done. The mixing height correction needs much better explanation. Even after reading Zhu et al. (2016) it is not clear how or if the Kh values were adjusted after the 40% mixing height correction. The fact that GEOS overpredicts mixing height so much indicates significant errors in surface fluxes or air temperature or winds which could adversely affect the AQ simulation. The emission adjustments are also not sufficiently explained or justified. Over predictions of NOx concentrations do not necessarily indicate emission over prediction, especially when the meteorology simulation has such large errors. The isoprene corrections are

not explained at all. I think that these adjustments are very questionable. The fact that you get better results is not sufficient justification. It is likely that you are adding errors to compensate for other errors.

P3 ln33: More needs to be said than just "simply explained by numerical diffusion". An AQ model that cannot get the high end of ozone distribution is not very useful.

P4 ln 7-10: Were model values for u*, L, Vd, Ra used in the 10 m calculations or the "typical" values given in the text? In any case, this technique has its limitations and uncertainties that should be noted since ozone is not an inert tracer. The ozone profile between 60m and 10m is affected not only by deposition flux but also chemical reactions with NOx and VOCs that usually have the opposite gradients from ozone. It would be preferable to run both the meteorology and chemical models at finer vertical resolution near the ground so that the model explicitly simulates 10m concentrations.

P4 ln4-18: The large underprediction of cloud cover is another significant deficiency in this model. This, combined with the large overpredictions of mixing height, suggest that meteorology model is not sufficiently realistic for modeling boundary layer air quality.

P7 ln1-2: This statement is way too strong considering the other errors in the model system!

P7 ln9: Is the grey shading defined by model or observations?

P7 ln 12-15: There is asymmetry in the Hotslag&Boville93 scheme for potential temperature. However, how this is applied to chemical concentrations is not explained.

P7 ln22-24: Removing the non-local term actually removes the asymmetry which contradicts the findings of Wyngaard and Brost and others.

Section 6: Another possible reason for the poor modeling of the ozone gradient could be that the model underpredicts the ozone concentrations above the mixing layer. On both days the ozone profile increases throughout the PBL and above the PBL. Having high concentration above the PBL to entrain would tend to increase concentrations in

the upper part of the PBL. The greater gradient from the "corrected" GEOS-Chem is simply due to decreased mixing from an arbitrarily reduced Kz profile.

---

## Referee Comment (RC2) · Anonymous Referee #1 · 9 Oct 2017

Travis et al. presented a comprehensive model and observation comparison of horizontal and vertical ozone distributions over Southeastern US. The comparison started from adjusting the NOX emission in the model framework as recent studies have suggested and moved to discussion on meteorological parameter and boundary layer mixing treatments in the GEOS-Chem model framework. They reported two issues in the model framework – low ozone from precipitation events were underestimated and vertical mixing, especially, a strong asymmetry between top-down and bottom-up mixing are ill-represented in the model framework. I am an experimentalist so have limited knowledge in modeling but I believe this work will remind the research community the importance of representation of the dynamics of boundary layer, which has not been a high priority in our research community. In this perspective, this manuscript will be
* * *
Interactive
comment

highly beneficial to suggest a new research direction. The manuscript in general is well written and easy to follow. I have two suggestions that may improve the clarity of the arguments.

1) I believe the research flights usually conduct spiral maneuverings and was wondering that the spiral profiles can be utilized the ozone the vertical distribution analysis presented in Section 6.

2) May be I missed something but the vertical ozone distribution with suppressed the top-down mixing presented in Figure 6 has substantial bias with observed ozone in the lower mixed layer in terms of values although the shape is reasonably simulated for the cloudy day simulation. Discussion on this would be beneficial.

---

## Author Comment (AC1) · 17 Nov 2017

*We thank the two reviewers for their comments. Reviewer comments are shown in red. Our responses are shown in blue, with new text in bold. Line numbers referenced here refer to the tracked-changes version of the document.*

**Anonymous Referee #1**

**1) I believe the research flights usually conduct spiral maneuverings and was wondering that the spiral profiles can be utilized the ozone the vertical distribution analysis presented in Section 6.**

This is sometimes true, but was not the case for SEAC$^4$RS. We added the following text on page 8 line 15, and page 12, lines 5-6 to address this comment.

**"Aircraft observations do not effectively probe that region of the atmosphere but ozonesondes do."**

**"and additional profile observations of the evolution of meteorological tracers, ozone and other long-lived chemical species in the boundary layer are essential to testing model parameterizations."**

**2) Maybe I missed something but the vertical ozone distribution with suppressed the top-down mixing presented in Figure 6 has substantial bias with observed ozone in the lower mixed layer in terms of values although the shape is reasonably simulated for the cloudy day simulation. Discussion on this would be beneficial.**

We added the following text on page 9, lines 27-28,

"These specific days have free tropospheric **and surface** biases but our interest here is in the simulation of the PBL vertical gradient **which is unaffected by these biases**."

And the following text on Page 10, lines 18-22,

**"The model underestimate of ozone above the mixed layer shown in Figure 6 cannot explain the missing model gradient because any additional entrained ozone will rapidly into a smooth vertical profile due to the fast mixing in the current scheme. Figure 5 shows several profiles where model ozone is overestimated above the mixed layer but the gradient below is unaffected."**

**Anonymous Referee #2**

**I think that the issue that this paper raises, the modeling of vertical gradients of ozone, is interesting and potentially important. However, there are too many shortcomings to the modeling to give a definitive analysis of the issue. An obvious shortcoming is that the lowest model level is 60 m while the measurements are at 10 m. Most other air quality models use much finer grid resolution near the surface.**

It is true that regional models have finer grid resolution near the surface, but this is not true of global models. CAM-Chem (Lamarque et al, 2012) has the same vertical levels as GEOS-Chem when using assimilated meteorology. We correct for the coarseness of our lowest model level in Section 3. We add the following text to page 3, line 30.

"**,typical for global CTMs (i.e. Lamarque et al, 2012)**"

**Furthermore, I strongly object to using different PBL schemes for meteorology and chemistry.**

We perform a sensitivity test using both schemes within the GEOS-5 GCM and find very little difference in the ozone vertical structure in the boundary layer, as expected. See lines 1-2 on page 9, and line 1 on page 10 : "We conducted a sensitivity on-line simulation in the GEOS-5 GCM using the GEOS-Chem chemical module (Long et al., 2015) and including the GEOS-5 PBL mixing scheme of Lock et al. (2000), but found the same excessive downward mixing of ozone as in the off-line GEOS-Chem."

**Not only has the PBL height been reduced for the chemistry but the Kz is also reduced and non-local term eliminated for the proposed correction. This is unjustified. If the same scheme does not give realistic results for both meteorology and chemistry, it should not be used for combined meteorology and air quality modeling.**

The mixing depths from GEOS are purely diagnostic, since mixing is done online in GEOS-5. The non-local term is actually calculated online We add the following clarifying text to page 3, line 32 and page 4 line 3,

"The parameterization uses **diagnostic** mixing depths …"
"**These diagnostic** mixing depths…"

and remove the reference on page 4, line 18 ("daytime mixing depths (Zhu et al, 2016) to avoid confusion.

**The other corrections and adjustments, such as reducing NOx emissions, that are made in order to get better results also call into question the validity of the study and its findings.**

The other corrections and adjustments, such as reducing NOx emissions, are strongly supported by Travis et al, 2016. See page 4, lines 23-28.

**P2 ln17-19: The statement: "we showed that the NOx National Emission Inventory (NEI) from the US Environmental Agency (EPA, 2015) was too high by 30-50 %." Is much too strong. All they should say is that this model with its many demonstrated errors better predicts ozone and NOx with a 30-40% emission cut. I do not believe that it's been proven that the NEI is over predicting NOx emissions by this amount**.

We softened this language throughout, including changing the text on page 1, line 19 "After correcting for a 30-50 % NOx emission overestimate in the US EPA National Emission Inventory…" to

**"Some of the model bias appears due to an overestimate of NOx emissions, and after correcting for this overestimate…"**

and removed the discussion starting with "One the basis of observations…" on page 2.

**P3 ln6-7: I don't understand how the mixing depth (h) can be defined by Kh values when according to Holtslag and Boville (1993) the Kh is a function of h which is defined by bulk Richardson number. Also, if the mixing depth is 40% too high, then this problem should be diagnosed and fixed. It is not reasonable to "correct" this in Geos-Chem. The result would be inconstancies between chemical concentration profiles and meteorological profiles which would lead to many errors in the chemical simulation including incorrect advection, errors in temperature and humidity especially above the GEOS-Chem h but below the GEOS h. I think that such "corrections" in AQ modeling should not be acceptable practice.**

The mixing depth from the GEOS-5 simulation is purely diagnostic, they do not represent how mixing in the GEOS-5 system is actually calculated online. We add the following text on page 3, line 32,

"The parameterization uses **diagnostic** mixing depths…"

**P3 ln20-22: More explanation and justification should be provided for the corrections and adjustments to PBL, emissions, and chemistry. The reader should not have to look up these other papers to know what was done.**

We suggest that this type of referencing is standard practice.

**The mixing height correction needs much better explanation. Even after reading Zhu et al. (2016) it is not clear how or if the Kh values were adjusted after the 40% mixing height correction. The fact that GEOS overpredicts mixing height so much indicates significant errors in surface fluxes or air temperature or winds which could adversely affect the AQ simulation.**

See response above, the mixing depth from GEOS provided to GEOS-Chem is diagnostic only.

**The emission adjustments are also not sufficiently explained or justified. Over predictions of NOx concentrations do not necessarily indicate emission over prediction, especially when the meteorology simulation has such large errors. The isoprene corrections are not explained at all.  I think that these adjustments are very questionable. The fact that you get better results is not sufficient justification. It is likely that you are adding errors to compensate for other errors.**

We remove references to the NOx overestimate and other previous work on page 2, lines 30-34, and page 3, lines 1-2, since it is not relevant to this study.

**P3 ln33: More needs to be said than just "simply explained by numerical diffusion". An AQ model that cannot get the high end of ozone distribution is not very useful.**

We removed the text "explained by numerical diffusion" and clarified our meaning on page 4, lines 24-25 by adding the following,

**"…attributed to spatial averaging in the model…"**

**P4 ln 7-10: Were model values for u\*, L, Vd, Ra used in the 10 m calculations or the "typical" values given in the text? In any case, this technique has its limitations and uncertainties that should be noted since ozone is not an inert tracer. The ozone profile between 60m and 10m is affected not only by deposition flux but also chemical reactions with NOx and VOCs that usually have the opposite gradients from ozone. It would be preferable to run both the meteorology and chemical models at finer vertical resolution near the ground so that the model explicitly simulates 10m concentrations.**

We changed to text on page 5, line 20 from We combine… to
**"For example,…"**
to clarify that this is an example calculation for illustration purposes. We also add the following clarification on page 5, lines 23-24 to address the comment about chemical reactions.

**"This assumes conservation of the vertical ozone flux in the 10-60 m column, a safe assumption since the transport time is only a few minutes."**

**P4 ln4-18: The large underprediction of cloud cover is another significant deficiency in this model. This, combined with the large overpredictions of mixing height, suggest that meteorology model is not sufficiently realistic for modeling boundary layer air quality.**

Page 7, line 11 – we explain that missing cloud is a typical model problem not unique to GEOS.

Climate models in general tend to underestimate low cloud cover (Zhang, 2005; Mueller et al., 2006; Chepfer et al., 2008; Naud et al., 2010; Kay et al., 2012; Nam et al., 2012).

**P7 ln1-2: This statement is way too strong considering the other errors in the model system!**

We changed the sentence on page 9, lines 19-20 "This confirms that the model overestimate of surface ozone …" To:

**"This confirms that a source of model bias in simulating surface …"**

**P7 ln9: Is the grey shading defined by model or observations?**

The caption of Figure 6 states: "The grey shading in the bottom left panel indicates the cloud vertical extent as diagnosed from the ozonesonde relative humidity measurement."

**P7 ln 12-15: There is asymmetry in the Hotslag&Boville93 scheme for potential temperature. However, how this is applied to chemical concentrations is not explained.**

**P7 ln22-24: Removing the non-local term actually removes the asymmetry which contradicts the findings of Wyngaard and Brost and others.**

We add the following to clarify that we are arguing against counter-gradient transport for subsiding air masses on page 10, line 10.

**"**Additional non-local **(counter-gradient)…"**

**Section 6: Another possible reason for the poor modeling of the ozone gradient could be that the model underpredicts the ozone concentrations above the mixing layer. On both days the ozone profile increases throughout the PBL and above the PBL. Having high concentration above the PBL to entrain would tend to increase concentrations in the upper part of the PBL. The greater gradient from the "corrected" GEOS-Chem is simply due to decreased mixing from an arbitrarily reduced Kz profile.**

We added the following clarifying text on Page 10, lines 15-19,

**"The model underestimate of ozone above the mixed layer shown in Figure 6 cannot explain the missing model gradient because any additional entrained ozone will rapidly into a smooth vertical profile due to the fast mixing in the current scheme. Figure 5 shows several profiles where model ozone is overestimated above the mixed layer but the gradient below is unaffected."**

*References*

Lamarque, J. F., L. K. Emmons, P. G. Hess, D. E. Kinnison, S. Tilmes, F. Vitt, C. L. Heald, E. A. Holland, P. H. Lauritzen, J. Neu, J. J. Orlando, P. J. Rasch and G. K. Tyndall (2012). "CAM-chem: description and evaluation of interactive atmospheric chemistry in the Community Earth System Model." Geoscientific Model Development **5**(2): 369-411.

Travis, K. R., D. J. Jacob, J. A. Fisher, P. S. Kim, E. A. Marais, L. Zhu, K. Yu, C. C. Miller, R. M. Yantosca, M. P. Sulprizio, A. M. Thompson, P. O. Wennberg, J. D. Crounse, J. M. St. Clair, R. C. Cohen, J. L. Laughner, J. E. Dibb, S. R. Hall, K. Ullmann, G. M. Wolfe, I. B. Pollack, J. Peischl, J. A. Neuman and X. Zhou (2016). "Why do models overestimate surface ozone in the Southeast United States?" Atmospheric Chemistry and Physics **16**(21): 13561-13577.

Zhu, L., D. J. Jacob, P. S. Kim, J. A. Fisher, K. Yu, K. R. Travis, L. J. Mickley, R. M. Yantosca, M. P. Sulprizio, I. De Smedt, G. González Abad, K. Chance, C. Li, R. Ferrare, A. Fried, J. W. Hair, T. F. Hanisco, D. Richter, A. Jo Scarino, J. Walega, P. Weibring and G. M. Wolfe (2016). "Observing atmospheric formaldehyde (HCHO) from space: validation and intercomparison of six retrievals from four satellites (OMI, GOME2A, GOME2B, OMPS) with SEAC4RS aircraft observations over the southeast US." Atmospheric Chemistry and Physics 16(21): 13477-13490.

**Resolving ozone vertical gradients in air quality models**

Katherine R. Travis[1], Daniel J. Jacob[1,2], Christoph A. Keller[3,4], Shi Kuang[5], Jintai Lin[6], Michael J. Newchurch[7], Anne M. Thompson[8], Jeff Peischl[9,10]

[1]School of Engineering and Applied Sciences, Harvard University, Cambridge, MA, USA
[2]Department of Earth and Planetary Sciences, Harvard University, Cambridge, MA, USA
[3]Universities Space Research Association, Columbia, MD, USA
[4]NASA Goddard Space Flight Center, Greenbelt, MD, USA
[5]Earth System Science Center, University of Alabama in Huntsville, Huntsville, AL 35805, USA
[6]Laboratory for Climate and Ocean-Atmosphere Studies, Department of Atmospheric and Oceanic Sciences, School of Physics, Peking University, Beijing 100871, China
[7]Atmospheric Science Department, University of Alabama in Huntsville, Huntsville, AL 35805, USA
[8]Earth Science Division, NASA Goddard Space Flight Center, Greenbelt, MD 20771, USA
[9]University of Colorado, Cooperative Institute for Research in Environmental Sciences, Boulder, CO, USA
[10]NOAA Earth System Research Lab, Boulder, CO, USA

*Correspondence to*: Katherine R. Travis (ktravis@fas.harvard.edu)

**Abstract.** Models severely overestimate surface ozone in the Southeast US during summertime and this overestimation has implications for the design of air quality regulations.. We use the GEOS-Chem model to interpret ozone observations over that region from aircraft (SEAC[4]RS), ozonesondes (SEACIONS), and surface sites (CASTNET) in August-September 2013. After correcting for a 30-50 % NO$_x$ emission overestimate in the US EPA National Emission Inventory,Some of the model bias appears due to an overestimate of NO$_x$ emissions, and after correcting for this overestimate 
[revised manuscript text omitted]

GEOS-FP data, defined as the GCM level above which the eddy diffusivity for heat falls below a threshold value of 2 m$^2$ s$^{-1}$ (McGrath-Spangler and Molod, 2014). These diagnostic mixing depths were found to be 40 % too high compared to the SEAC$^4$RS aerosol lidar data and were reduced accordingly for application of the Holstag and Boville (1993) parameterization in GEOS-Chem (Zhu et al., 2016).

Ozone deposition in GEOS-Chem follows the resistance-in-series scheme of Wesely (1989) as implemented by Wang et al. (1998) and further modified for SEAC$^4$RS conditions by Travis et al. (2016). The mean midday ozone deposition velocity over the Southeast US in the model is 0.8 ± 0.3 cm s$^{-1}$ during August-September 2013. Comparison with ozone deposition measurements by Finkelstein et al. (2000) at Duke Forest, North Carolina shows good agreement with a mean ozone deposition velocity of 0.8 cm s$^{-1}$ during daytime. Aircraft eddy covariance flux measurements over the Ozarks forest during SEAC$^4$RS indicate a daytime ozone deposition velocity of 0.8 ± 0.1 cm s$^{-1}$, in agreement with the local GEOS-Chem value of 0.9 cm s$^{-1}$ (Wolfe et al., 2015).

Detailed evaluations of GEOS-Chem with SAS and SEAC$^4$RS observations have been reported in previous studies. Initial evaluations led to corrections of NEI NO$_x$ emissions (Travis et al., 2016) and isoprene chemistry (Fisher et al., 2016; Travis et al., 2016). After these corrections, the model was found to be successful in reproducing surface and aircraft observations of aerosol composition (Kim et al., 2015b; Marais et al., 2016) and organic nitrates (Fisher et al., 2016), and aircraft observations of formaldehyde (Zhu et al., 2016), glyoxal (Miller et al., 2017), and ozone and its precursors (Travis et al., 2016; Yu et al., 2016). Travis et al. (2016) presented successful model comparisons to observations of (1) NO$_x$, (2) the relationship of ozone to NO$_x$ oxidation products (a measure of the ozone production efficiency), and (3) isoprene nitrates and peroxides tracking the high-NO (ozone-producing) and low-NO pathways for isoprene oxidation. This evaluation lends some confidence in the model simulation of ozone chemistry.

**3   Ozone frequency distributions in the mixed layer and surface air**

Figure 1 (left panel) shows the frequency distribution of afternoon (12-18 local time) ozone concentrations in August-September 2013 measured by the SEAC$^4$RS DC-8 aircraft in the mixed layer at 0.4-1.0 km altitude. The mean ozone in the mixed layer as measured by the aircraft is 50 ± 10 ppb. The model sampled along the aircraft tracks is in good agreement (52 ± 10 ppb, $r$=0.54). The model does not capture the

observed extremes and this can be simply attributed to spatial averaging in the model (Yu et al., 2016) and targeted sampling by the aircraft. In particular, observations above 75 ppb are associated with focused sampling of urban (Houston) and agricultural fire plumes that may not be properly represented or located in the model (Travis et al., 2016).

Also shown in Figure 1 (right panel) is the frequency distribution of maximum daily 8-hour average (MDA8) ozone at the CASTNET regional air quality surface network for the same period (https://www.epa.gov/castnet).  The mean MDA8 ozone measured at CASTNET sites is 40 ± 9 ppb, while the corresponding model mean sampled at the lowest grid level is 48 ± 9 ppb, for a high mean bias of 8 ± 9 ppb.

Part of the  bias in the model comparison to  the CASTNET data can be corrected by taking into account the vertical gradient between the lowest model grid level (60 m  and the 10-m altitude at which the CASTNET measurements are typically made. This gradient is effectively implied by the local model aerodynamic resistance ($R_a$) used to compute ozone dry deposition, and it is readily extracted from the model output (Zhang et al., 2012). For example, a typical friction velocity $u* = 0.4$ m s$^{-1}$ and daytime Monin-Obhukov length $|L| = $ 100 m yields $R_a = 0.07$ s cm$^{-1}$ between 60 and 10 m; combining this with an ozone deposition velocity of 0.8 cm s$^{-1}$ implies an ozone decrease of approximately 3 ppb between the two altitudes. This assumes conservation of the vertical ozone flux in the 10-60 m column, a safe assumption since the transport time is only a few minutes.

[revised manuscript text omitted]

Wyngaard and Brost (1984) used large-eddy simulations to investigate top-down vs. bottom-up differences in eddy diffusion parameterizations of PBL mixing. They show that eddy diffusion coefficients ($K_z$) for top-down transport should be about 60 % lower than for bottom-up transport, due to the role of surface-driven buoyant plumes in contributing to bottom-up transport. Additional non-local (counter-gradient) vertical transport in PBL schemes, developed originally for heat flux, is mostly intended to resolve buoyant plumes (Deardorff, 1966; Holtslag and Moeng, 1991) and should be formulated differently for top-down transport (Xie and Fung, 2014). We conducted a sensitivity simulation  where the Holtslag and Boville (1993) mixing scheme was modified for ozone to decrease $K_z$ by 60 % and remove the non-local term. As shown in Figure 6, this fully corrects the ozone gradient on the two sample days of Figure 6. The model underestimate of ozone above the mixed layer shown in Figure 6 cannot explain the missing model gradient because any additional entrained ozone will rapidly into a smooth vertical profile due to the fast mixing in the current scheme. Figure 5 shows several profiles where model ozone is overestimated above the mixed layer but the gradient below is unaffected.

The need for asymmetric top-down vs. bottom-up PBL mixing for air quality applications has long been recognized (Pleim and Chang, 1992), and is presently implemented in the EPA Community Multiscale Air Quality (CMAQ) and in the Comprehensive Air quality Model with Extensions (CAMx) using the Asymmetrical Convection Model version 2 (ACM2) (Pleim, 2007a, b). The ACM2 has the same eddy diffusion component as Holtslag and Boville (1993) but a different form of nonlocal parameterization. It treats upward convective transport with a nonlocal buoyant component, but downward transport as a slower, layer-by-layer process. However, comparisons to ozonesonde and aircraft observations suggest that ACM2 still has excessive mixing for ozone down to the surface (Tang et al., 2011; Goldberg, 2015).

**7 Conclusions**

Models overestimate summertime surface ozone in the Southeast US. Some of this bias may be attributed to an overestimate of $NO_x$

emissions in the US EPA National Emission Inventory (Travis et al., 2016). However, midday ozonesondes also show a large vertical gradient of decreasing ozone below 1 km altitude that is at odds with the strong mixing expected from models. Here we investigated the cause of this discrepancy through the combined analysis of August-September 2013 ozone observations from aircraft (SEAC[4]RS), surface (CASTNET), and ozonesondes (SEACIONS).

Statistical comparison of the GEOS-Chem model to aircraft observations of ozone below 1 km shows no significant bias (50 ± 10 ppb observed, 52 ± 10 ppb model), but the maximum daily 8-h average (MDA8) surface ozone at CASTNET sites is overestimated by 8 ± 9 ppb (40 ± 9 ppb observed, 48 ± 9 ppb model).  Part of that discrepancy is simply due to a subgrid ozone gradient between 60 m altitude (lowest model grid level ) and 10 m (where the measurements are typically made). Increasing vertical grid resolution is not necessary as a subgrid correction can be readily applied using the model aerodynamic resistance to dry deposition under the assumption of uniform vertical flux. This correction, which is generally ignored in models, averages 3 ppb in our case. The resulting model ozone at 10 m altitude is 45 ± 8 ppb, still significantly higher than observed. August-September 2013 was cooler and wetter than average but the effect on ozone was small, averaging 2 ppb at CASTNET sites. The low tail of observed MDA8 ozone (<25 ppb) was largely associated with rainy conditions.

The GEOS-FP meteorological data driving GEOS-Chem are biased toward clear-sky, and this bias would be expected to contribute to the overestimate of ozone. However, we find that the model MDA8 ozone is only 4 ppb lower under low-cloud and rainy conditions than in clear sky, whereas in the observations that difference is 7 ppb under low-cloud conditions and 11 ppb under rainy conditions. Midday ozonesonde data from Huntsville, Alabama show a 6 ppb decrease from 1 km to the surface (4 ppb under clear-sky, 7 ppb under low cloud), whereas the model shows only a 1 ppb decrease. Thus the model has excessive top-down mixing of ozone; this is seen using both the Holtslag and Boville (1993) PBL scheme in the off-line GEOS-Chem and the Lock et al. (2000) scheme in the GEOS-5 GCM. By contrast, potential temperature shows similar strong vertical mixing in the observations and the model. Bottom-up mixing (as for heat) is known to be faster than top-down mixing (as for ozone) because of buoyant plumes but the two above schemes do not include this asymmetry. The ACM2 scheme (Pleim, 2007b, a) includes this asymmetry, but previous evaluations suggest that this scheme still has excessive downward mixing of ozone. We find in a sensitivity simulation that decreasing top-down eddy diffusion following Wyngaard and Brost (1984) and suppressing top-down non-local vertical transport allows GEOS-Chem to successfully simulate the observed ozone gradient in the mixed layer. More work is needed to describe the top-down mixing of

ozone for air quality applications. and additional profile observations of the evolution of meteorological tracers, ozone and other long-lived chemical species in the boundary layer are essential to testing model parameterizations.

**8 Data availability**

Cloud data from the Automated Surface Observing System (ASOS) can be downloaded here: http://mesonet.agron.iastate.edu/request/download.phtml. PRISM temperature and precipitation data can be downloaded here: http://www.prism.oregonstate.edu/historical/. The SEACIONS ozonesonde data can be accessed here: https://tropo.gsfc.nasa.gov/seacions. The CERES cloud fraction and cloud optical depth observations are available at http://doi.org/10.5067/Aqua/CERES/ISCCP-D2LIKE-MERG00_L3.003. The SEAC⁴RS aircraft data can be found here: https://www-air.larc.nasa.gov/missions/seac4rs/DC8-Extract.html. CASTNET data are available at here: https://www.epa.gov/castnet.

**9 Competing Interests**

The authors declare that they have no conflict of interest.

*Acknowledgements*

We thank Randal Koster (NASA), Dan Goldberg (ANL), and Taylor Jones, Eloise Marais, Rachel Silvern, and Lu Shen (Harvard) for helpful discussions. This work was supported by the NASA Earth Science Division. AMT acknowledges SEACIONS support from the Tropospheric Chemistry Program to NASA/Goddard, NOAA/ESRL/GMD and originally to Penn State University (Grant NNX12AF05G). We thank Tom Ryerson (NOAA), Jeff Peischl (NOAA), and Ilana Pollack (CSU) for use of their ozone measurements from the NOAA NOyO3 instrument.

[Figure]

**Figure 1** – Probability density functions (pdfs) of ozone concentrations in the Southeast US (94.5-80 W, 29.5-38 N, maps inset with sampling locations indicated) in August-September 2013. Mean and standard deviation are given in the legend for each pdf. The left panel shows afternoon (12-18 local time) mixed layer values measured by the SEAC[4]RS DC8 aircraft at 0.4-1.0 km altitude (*n* = 370). The right panel shows maximum 8-hour daily average (MDA8) near-surface values (about 10 m above the local surface) measured at the CASTNET network of 15 rural sites. Also shown are the corresponding GEOS-Chem model pdfs sampled at the locations and times of the observations. The thin red line in the right panel is the model pdf for the lowest model level (centered at 60 m above ground). The thick red line is the implied model value at 10 m above ground (see text).

[Figure]

[Figure]

**Figure 2** – Ozone and weather variables averaged over the 15 Southeast US CASTNET sites of Figure 1, and over August-September 1987-2015. The top panel shows the  trend  and linear regression  in MDA8 ozone. The bottom panel shows mean daily temperature ( average of daily minimum and maximum temperatures) and precipitation  from the PRISM Climate Group datasets (http://www.prism.oregonstate.edu). Dashed lines indicate the 1987-2015 mean values. Circles highlight 2013.

[Figure]

**Figure 3** – Probability density functions (pdfs) of MDA8 ozone at CASTNET sites in the Southeast US in August-September 2013. The pdfs are segregated between clear-sky, low cloud, and rainy conditions as described in Section 4. The model pdfs include the correction for 10 m ozone described in Section 3. For each sky condition, the mean ozone and its standard deviation are given inset with the frequency of that sky condition in parentheses. The frequencies do not add up to 100 % because partial low-cloud cover (0.5-3 oktas) is not included.

[Figure]

**Figure 4** – Average daytime low-cloud fraction (below 680 hPa, 9-17 local time) in August-September 2013. The left panel shows satellite data from the CERES ISCCP-D2like product (CERES Science Team, Hampton, VA, USA: NASA Atmospheric Science Data Center, accessed May, 2016, at http://doi.org/10.5067/Aqua/CERES/ISCCP-D2LIKE-MERG00_L3.003A). This merged product combines 3-hourly, daytime cloud properties from Terra and Aqua on the Moderate Resolution Imaging Spectroradiometer (MODIS) and geostationary meteorological satellites mapped on a 1°×°×1° grid (Minnis et al., 2011). The right panel shows data from GEOS-FP, where cloud fraction and in-cloud optical depth are provided for each model level, using the maximum random overlap scheme (MRAN) to derive total cloudiness below 680 hPa (Liu et al., 2006).

**Table 1 -** CERES and GEOS-FP low-cloud frequencies in the Southeast US.[1]

|  | CERES Low-Cloud | | GEOS-FP Low-Cloud | |
|---|---|---|---|---|
|  | **Fraction** | **Optical Depth** | **Fraction** | **Optical Depth** |
| **Cumulus** | 11% | 1.6 | <1% | 1.3 |
| **Stratocumulus** | 9% | 8 | 6% | 13 |
| **Stratus** | 1% | 36 | 3% | 31 |

[1]Data from August-September 2013 for the domain of Figure 4. The classification of low-cloud type is done by CERES according to optical depth below 680 hPa: cumulus (0.02-3.55), stratocumulus (3.55-22.63), and stratus (22.63-378.65).

[Figure]

[Figure]

**Figure 5** – Midday vertical profiles of ozone over Huntsville, Alabama (35.3 N, 86.6 W) for the full troposphere (up to 12 km, top) and for the PBL (up to 3 km, bottom). Ozonesonde observations (*n* = 31 during 08 August – 21 September 2013, launched at 10-13 local time) are compared to GEOS-Chem model profiles sampled at the same location and times. Values are interpolated in time between launches and are not intended to resolve the diurnal cycle of ozone. The ASOS low-cloud fraction at the time of the ozonesonde launch and daily PRISM precipitation (mm d$^{-1}$) are also shown along with the corresponding model values. Clear, low-cloud, and rainy days following the criteria of Section 4 are labeled in color in the abscissa. The black diamonds on the bottom plot show midpoints of the model grid levels.

[Figure]

**Figure 6 -** Vertical profiles of ozone concentrations and potential temperature at the SEACIONS Huntsville site on representative clear-sky and low-cloud days from the record of Figure 5. The left panels include the sensitivity simulation with reduced top-down mixing in the mixed layer as described in Section 5. The grey shading in the bottom left panel indicates the cloud vertical extent as diagnosed from the ozonesonde relative humidity measurement.